# The adaptive large language models for vaccine prediction: A novel approach to vaccine demand prediction with engineered deviation prompts

Mingming Chen[1,2], Qiangsong Wu[3], Zilong Wang[4], Qi Qi[1], Yi Hu[5]*, Tenglong Li[1,2]*

**1** Academy of Pharmacy, Xi'an Jiaotong-Liverpool University, Ren'ai Road, Suzhou, Jiangsu, People's Republic of China, **2** Institute of Population Health, Faculty of Health & Life Sciences, Waterhouse Building, University of Liverpool, Liverpool, United Kingdom, **3** Department of Immunization Program, Xuhui District Center for Disease Control and Prevention, Shanghai, People's Republic of China, **4** Department of Biostatistics, T.H. Chan School of Public Health, Harvard University, 655 Huntington Avenue, Building 2, Boston, Massachusetts, United States of America, **5** School of Public Health and Key Laboratory of Public Health Safety, Fudan University, 130 Dong'an Road, Xuhui, Shanghai, People's Republic of China

* yhu@fudan.edu.cn (YH); Tenglong.Li@xjtlu.edu.cn (TL)

## Abstract

Accurate vaccine demand forecasting is crucial for minimizing wastage and ensuring efficient immunization programs. In this study, we introduce an Adaptive Large Language Model for Vaccine Prediction (ALLMVP) that integrates large language model (LLM) architectures with an adaptive value correction mechanism. Using vaccination record data from Xuhui District, Shanghai, China (2014–2022), we conducted a comparative analysis of ALLMVP against seven other models, including standard machine learning methods (logistic regression, random forest, Long Short-Term Memory) and their enhanced versions with the same adaptive value correction mechanism. Our findings indicate that traditional models encountered significant challenges in attaining high predictive accuracy, while frameworks based on LLMs markedly enhanced their forecasting capabilities. Notably, ALLMVP achieved an acceptance ratio of 71.43%, which is considerably superior to that of the other models under consideration. Yearly and cumulative evaluations across seven vaccines from the National Immunization Program demonstrated that ALLMVP consistently delivered more precise estimates, aligning closely with actual vaccine demand even under challenging conditions, such as the post-COVID era. These results highlight the potential of adaptive LLM-driven forecasting tools to fullfill stringent prediction accuracy standards set by governments and to aid in data-informed vaccination strategizing. The AI infrastructure underpinning ALLMVP holds the promise of being generalized and deployed across a range of forecasting applications and at a significantly larger scale.

**Data availability statement:** The data cannot be made publicly available due to the policy constraints on vaccination records established by Shanghai Center for Disease Control and Prevention. Individuals who are interested in getting the dataset can contact the Data Access Committee of Xuhui District CDC via xuhuimianyi@126.com for their specific requests.

**Funding:** The author(s) received no specific funding for this work.

**Competing interests:** The authors have declared that no competing interests exist.

## Author summary

Making life-saving vaccines available exactly when and where they are needed is a major challenge for public health. Inaccurate forecasts often lead to wasteful surpluses or dangerous shortages. In this study, we developed a new artificial intelligence tool, called ALLMVP, to bridge this gap. By combining the reasoning power of Large Language Models with a specialized mechanism that learns from past errors, our model predicts vaccine demand with much higher precision than traditional methods. We tested our approach using nearly a decade of vaccination records from Shanghai, China. Even during the unpredictable periods following the COVID-19 pandemic, our AI tool consistently provided estimates that closely matched actual usage, significantly outperforming other forecasting tools. We believe our research offers a practical solution for governments to improve their immunization programs and reduce waste. Beyond vaccines, the framework we built has the potential to be used for forecasting a wide range of medical supplies on a much larger scale, helping healthcare systems achieve balance between supply and demand.

## 1. Introduction

In China, vaccination has played a vital role in disease prevention. Throughout the interval spanning from 1978 to 2018, the incidence of major vaccine-preventable diseases experienced a substantial decline. For instance, the incidence of pertussis diminished from 126.35 to 1.58 cases per million (a 98% reduction), while measles, epidemic meningitis, and Japanese encephalitis exhibited declines of 99%, 99%, and 98%, respectively. Consequently, the aggregate life expectancy associated with these diseases increased by 0.79 years [1]. Such remarkable achievements are closely associated with both the persistent expansion of the National Immunization Program (NIP) vaccines in China—from four types in 1978 to fourteen types in 2008—and the high coverage rates [1–2]. These vaccines are provided at no cost by the government and must be administered in strict accordance with national regulations [3]. Annually, the National Administration of Disease Control and Prevention, in collaboration with the Ministry of Finance, centralizes vaccine procurement based on the usage plans submitted by provincial Centers for Disease Control and Prevention. The usage of NIP vaccines is influenced by numerous factors, including variations in birth rates, population mobility, the substitution rate of non-NIP vaccines, vaccine availability, and special events [4]. Study has demonstrated that the overall planned utilization rate of NIP vaccines (i.e., the ratio of distributed doses to planned usage) in Fujian Province from 2019 to 2021 fluctuated from 68% to 76%. The planned utilization rate exhibits considerable variability across different NIP vaccines, implying that substantial surpluses or shortages may materialize. For example, the planned utilization rate for the Group A meningococcal vaccine was merely 43% in 2020, in stark contrast to the measles-mumps-rubella (MMR) vaccine, which achieved a planned utilization rate of

134% [5]. It's imperative to avert vaccine surpluses that result in economic losses due to limited shelf life [6–7], as well as vaccine shortages that precipitate delays in vaccinations and consequently heighten health risks [8–10]. Therefore, accurately forecasting the NIP vaccine actual usage in the forthcoming year remains a significant objective for public health authorities.

In order to achieve the goals delineated in the Immunization Agenda 2030, which is predicated upon the World Health Organization (WHO) Global Vaccine Action Plan and the Healthy China 2030 Planning Outline, the deployment of artificial intelligence (AI) applications has been extensively utilized for the optimization of vaccine supply chain management to enhance the efficacy and cost-effectiveness of vaccination campaigns [11–15]. Previous studies have provided deep insights into constructing intricate models for vaccine predictions [16–17]. Nevertheless, the implementation of artificial intelligence (AI) within this domain encounters several challenges, which restrict the operational efficiency of these models in particular contexts [18–19]. First, the intricate nature of vaccine supply and demand dynamics obstructs AI models from producing precise forecasts across heterogeneous populations [20–21]. Furthermore, AI models are often characterized by a deficiency in robustness and generalizability, rendering them inadequate in fulfilling the diverse quality benchmarks established by various governmental entities [22–23]. Second, vaccination record data frequently exhibits irregular temporal patterns and is generally accompanied by a limited array of relevant features, thereby complicating the training processes of AI models. The Management of small feature sets with long irregular timer series presents a significant challenge, especially for methodologies such as decision trees and random forests, which may encounter difficulties when addressing complex or nonlinear relationships [24]. Third, the inherent lack of transparency and explainability associated with numerous machine learning models acts as a barrier to their integration within public health frameworks. Health professionals require a clear comprehension of the predictive processes to cultivate trust and facilitate the implementation of AI-driven recommendations. Enhancing the interpretability of these models could improve their applicability by facilitating a better understanding of predictions and limitations [25–26]. Due to these challenges, there exists a relative paucity of effective AI applications within the vaccine sector as documented in the extant literature. Addressing these difficulties is imperative for the progression of AI-driven solutions that can substantially advance vaccine research and bolster public health initiatives.

Traditional AI methodologies have shown limitations in adaptability for handling vaccination record data and yielding satisfactory predictions. To address these challenges, our study employs large language models (LLMs) for vaccine prediction, motivated by their inherent advantages in temporal analysis. Unlike conventional models, the core architecture of LLMs, such as self-attention mechanisms, enables more flexible capture of long-range dependencies within the data. Furthermore, their strong few-shot learning capabilities and prompt-based framework allow for effective integration of complex information and adaptation to limited data scenarios. These theoretical strengths position LLMs to achieve significantly superior performance compared to traditional methods [27–28]. Leveraging the architecture of LLMs, we can potentially predict vaccine demand with higher accuracy and thereby reduce the chance of vaccine wastage. In this paper, we propose the large language model for vaccine prediction (LLMVP) by integrating machine learning and the LLM technology. LLMVP is able to mine temporal patterns of vaccination records more deeply, thereby forecasting vaccine demand with higher accuracy and enabling better vaccine supply management. In addition, we propose the adaptive large language model for vaccine prediction (ALLMVP) by adding a value correction step to LLMVP, so as to adjust biases in LLMVP outputs and meet the quality standard for prediction (i.e., 5% margin of error) required by government. The predictive performances of LLMVP and ALLMVP are compared with other machine learning models such as logistic regression, random forest and long short-term memory (LSTM), as well as their enhanced versions with the adaptive value correction mechanisms. We aim to address the following two research questions based on our evaluation metrics: 1-whether large language model can yield predictions with higher accuracy than other models; 2-whether the adaptive value correction mechanism can further enhance predictive accuracies for the models in consideration.

## 2. Methods

### 2.1. Ethics statement

This study received ethical approval from the Xi'an Jiaotong-Liverpool University Research Ethics Review Panel (approval number: ER-LRR-11000026020241230162923).

### 2.2. Data

Using the Shanghai Immunization Program Information Management System (SIPIMS), we collected vaccination records for children aged 14 years or younger in Xuhui District, Shanghai, from 2014 to 2022 through a cluster sampling approach. A "vaccination record" refers to a record documenting the administration of a vaccine dose to an individual on a specific day at a specific clinic. If an individual receives multiple doses (different types of vaccines) on the same day, each dose is then saved as a distinct record. The data from 2014 to 2017 is used as training data, while the data from 2018 to 2022 is used as testing data for our analysis. A rolling-origin scheme was adopted for model training; the specific implementation details are provided in S3 Appendix.

The original dataset was cleaned to remove sensitive information and has the following variables: date of birth, sex, type of household registration (local or non-local), vaccination date, type of vaccine, dose of vaccine, code of vaccine (a unique identifier), product abbreviation, vaccine batch number, vaccine expiration date, vaccine price, and the name of the vaccination clinic. In particular, the variable 'code of vaccine' not only identifies the administered vaccine but also indicates whether it is part of the National Immunization Program (NIP) or not, with codes ending in 'A' typically denoting NIP vaccines and 'B' denoting non-NIP vaccines. Each NIP vaccine ID represents a unique vaccine, as outlined in Table 1.

### 2.3. Adaptive large language model for vaccine prediction

As illustrated in Fig 1, the proposed architecture, i.e., the Adaptive Large Language Model for Vaccine Prediction (ALLMVP), has two main components: the Large Language Model for Vaccine Prediction (LLMVP) and an error correction mechanism (See S1 Appendix for the pseudo-code.). During data preprocessing, we took several steps to ensure the dataset was clean and suitable for effective analysis. Data was checked for duplicate records, which were then removed to avoid result skewing. Additionally, data was normalized to maintain consistency across different scales and variables. The aforementioned preprocessing steps are crucial for ensuring output quality from AI models like GPT-4 [29–30]. The following two structured prompts were engineered to generate summaries for vaccine distribution (See prompt design in S1 Appendix):

- **Daily Vaccine Summary**: Records daily vaccine usage from 2018 to 2022, along with a daily average.

- **Monthly Vaccine Summary**: Captures monthly vaccine usage over the same period, summarized by monthly averages.

**Table 1. National Immunization Program (NIP) Vaccine List.**

| NIP vaccine ID | Vaccine name |
|---|---|
| A1 | Hepatitis B vaccine |
| A3 | Polio vaccine |
| A4 | Diphtheria, Tetanus, and Pertussis vaccine |
| A5 | Meningococcal vaccine |
| A6 | Measles, Mumps and Rubella (MMR) vaccine |
| A7 | Japanese encephalitis vaccine |
| A8 | Hepatitis A vaccine |

(NIP = National Immunization Program)

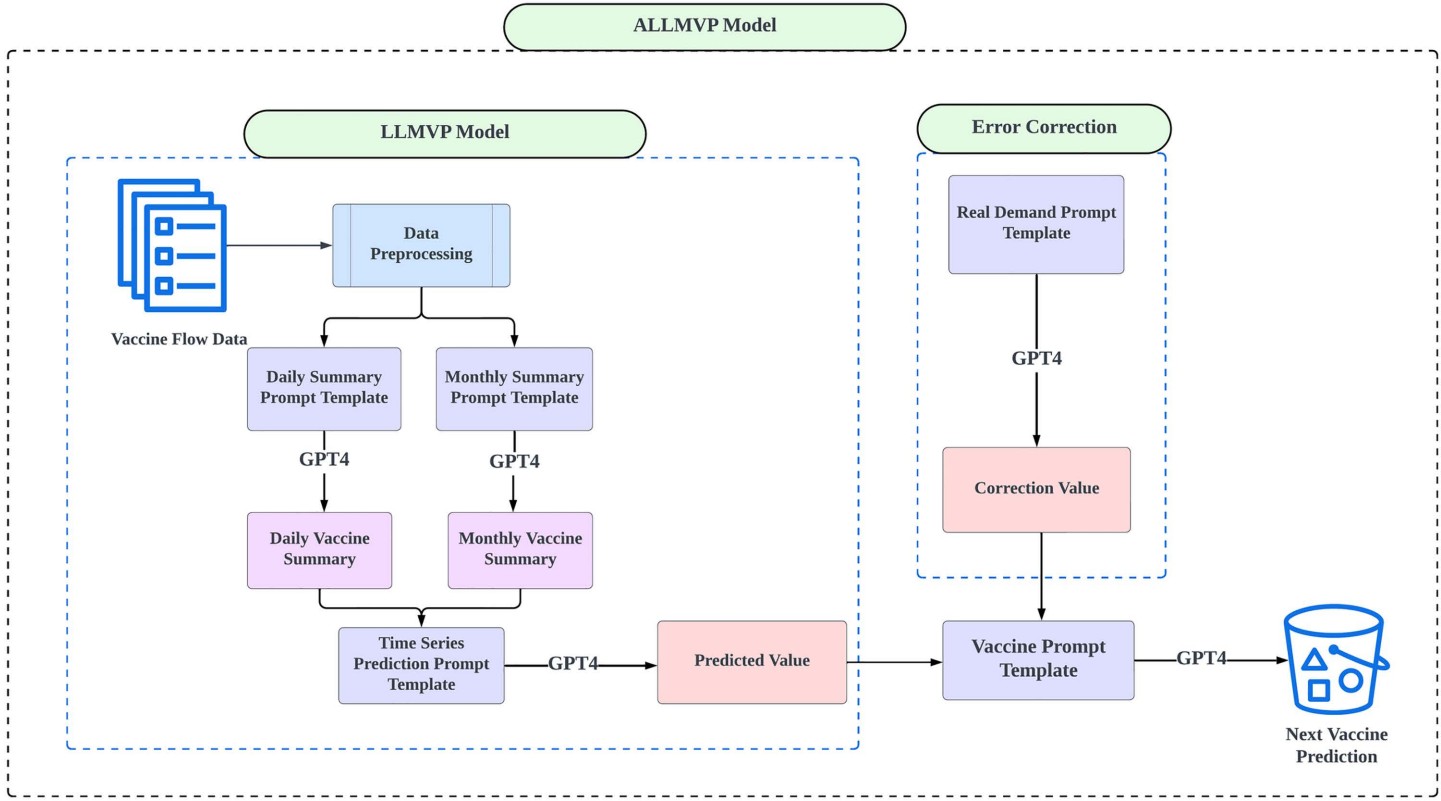

**Fig 1. Schematic of the ALLMVP framework.**

Equations 1 and 2 describe the methods for summarizing daily and monthly vaccine quantities, respectively. (See parameter descriptions in Table 2)

$$RN_{v,t_d} = DSummarize(P_{v,t_d}) \tag{1}$$

$$RN_{v,t_m} = MSummarize(P_{v,t_m}) \tag{2}$$

These summaries are merged for designated time intervals to provide additional inputs based on the data, for engineering the final time series prompts. Subsequently, utilizing the time series prompt template, GPT-4 is asked to compute the predicted value defined by Equation 3. (See parameter descriptions in Table 2)

$$PN_{v,t_y} = \Delta Predict(RN_{v,t_d},\ RN_{v,t_m}) \tag{3}$$

To further enhance model accuracy, we introduced an additional adaptive "value correction" step, which involves using real-time demand prompts to identify and correct deviations falling outside acceptable thresholds. This step is served as refinement of the original LLMVP architecture through effective correction of biases, thereby optimizing cost efficiency of the entire ALLMVP model. This adaptive correction step is activated when predicted values exceed a designated threshold, and thus it ensures values of low accuracy are not used in subsequent steps.

**Table 2. Definition of model parameters and functions.**

| Parameter | Description |
| --- | --- |
| $RN_{v,t_d}$ : | Real quantity of vaccine v on day $t_d$. |
| $RN_{v,t_m}$ : | Real quantity of vaccine v in month $t_m$. |
| $P_{v,t_d}$ : | Vaccinations administered to an individual for vaccine v on day $t_d$. |
| $P_{v,t_m}$ : | Vaccinations administered to an individual for vaccine v on month $t_m$. |
| DSummarize() : | Function that utilizes the GPT4 model to summarize daily vaccination occurrences. |
| MSummarize() : | Function that utilizes the GPT4 model to summarize monthly vaccination occurrences. |
| $PN_{v,t_y}$ : | Predicted quantity of vaccine v for the year $t_y$ using the LLMVP model. |
| $\Delta$Predict() : | The prediction operation using the GPT4 model. |
| $D_{t_y}$ : | List of correction deviations for the year $t_y$ |
| $\Delta$DPredict() : | The prediction operation using the GPT-4 model after applying corrections. |
| $DPN_{v,t_y}$ : | Corrected predicted quantity of vaccine v for the year $t_y$ |

Ultimately, the prediction and correction values are consolidated into the final vaccine prediction template, where correction values returned by the LLM are used to further calibrate the predictions given by the LLMVP model. This integrated template is designed to generate the final predicted vaccine quantities for the specified year range, as delineated in Equation 4. (See parameter descriptions in Table 2)

$$DPN_{v,t_y} = \Delta DPredict(D_{t_y},\ PN_{v,t_y}) \tag{4}$$

The integration of LLMVP and the additional adaptive value correction step gives GPT-4 more control for data with a desired threshold, and therefore it can increase the chance of acceptance stipulated by government compared to the original LLMVP for predicting vaccine demand.

## 2.4. Models in comparison

To evaluate the performance of our proposed Adapted Large Language Model for Vaccine Prediction (ALLMVP), we compare it against several predictive algorithms, which include traditional models and their enhanced versions with Large Language Models (LLMs). In total, eight models are adopted for comparison (See model details in S2 Appendix):

(1) Pure Logistic Regression (LR): A statistical method that models the relationship between a dependent variable and one or more independent variables using a logistic function [31].

(2) Random Forest (RF): An ensemble learning method that constructs multiple decision trees during training and outputs the mean prediction of the individual trees, suitable for handling complex, nonlinear relationships [32].

(3) Long Short-Term Memory (LSTM): A type of recurrent neural network capable of learning order dependence in sequence prediction problems, making it effective for time series data [33].

(4) Large Language Model for Vaccine Prediction (LLMVP): An initial predictive model leveraging GPT-4 to forecast vaccine demand based on time series data using structured prompts.

Building upon the above models, we introduce a value correction mechanism to enhance their predictive accuracy. This involves adding a correction value calculated using GPT-4, based on the differences among historical data, current prediction, and observed values. The correction value is used to adjust for biases and errors exceeding predetermined

thresholds, to meet the quality standard set up by government, i.e., annual vaccine demand forecasts should be within 5% margin of the actual values. The enhanced models are (See the model architecture in S1 Fig through S3 Fig):

(5) LR with LLM Correction (A-LR): Logistic Regression adjusted with the GPT-4 computed correction value.

(6) RF with LLM Correction (A-RF): Random Forest adjusted with the GPT-4 computed correction value.

(7) LSTM with LLM Correction (A-LSTM): LSTM adjusted with the GPT-4 computed correction value.

(8) Adaptive Large Language Model for Vaccine Prediction (ALLMVP): The LLMVP model adjusted with the GPT-4 computed correction value.

The correction mechanism is motivated by limitations in the traditional models, which focus solely on time series prediction and lack the ability to adapt for significant errors. For example, the LLMVP model has substantial biases when processing time series data, often yielding predictions outside the acceptable range. Considering that different regions may have their own vaccination policies and thus different expectations/thresholds (especially in some big cities that adopt stricter standards), there is a pressing need to refine predictive models to accommodate these variations effectively. Specifically, GPT-4 is employed to compute the correction value which is then integrated into the initial prediction to produce a corrected prediction more likely to fall within the acceptable threshold. The correction does not alter the original model architectures and is served to optimize the model outputs, particularly when predictions fail to meet the predetermined thresholds.

The following evaluation metrics are used for model comparisons:

- Acceptance Ratio: forecast falling within ±X% of the realized demand. In this study, the threshold was set at ±5%, as determined by the procurement tolerance range specified by the Shanghai Municipal Center for Disease Control and Prevention (CDC).

- Deviation Ratio: The average absolute deviation of the predicted values from the true values, normalized by the true values. It provides a measure of relative predictive accuracy.

- Root Mean Square Error (RMSE): A standard metric that measures the magnitude of the prediction errors by calculating the square root of the average squared differences between the predicted and observed values.

By comparing the above traditional and enhanced models, we can evaluate the effectiveness of the adaptive value correction mechanism in terms of accuracy improvement. The results from the model comparisons are presented next.

## 3. Results

To assess the accuracy and reliability of vaccine quantity prediction models, we conducted a comparative analysis of eight different algorithms' prediction accuracy for seven vaccines included in the National Immunization Program (NIP) between 2018 and 2022. In evaluating the performance of these vaccine demand forecasting algorithms, we employed three key metrics: Acceptance Ratio, Deviation Ratio, and Root Mean Square Error (RMSE) [34]. These metrics are details in the Table 3 for the models in comparison.

According to the results presented in Table 3, the comparison of individual model performance reveals distinct differences between the algorithms. Among the traditional models LR, RF and LSTM demonstrate relatively low prediction accuracy, with acceptance ratios of 17.14%, 17.14%, and 11.43%, respectively. These models show considerable deviation from the actual values, as evidenced by their high deviation ratios and root mean square error (RMSE) values. In contrast, the large language model for vaccine prediction (LLMVP), shows significant improvements across all three evaluation metrics (acceptance ratio: 28.57% versus 15.24%; deviation ratio: 0.1325 versus 0.2067; RMSE: 480.1619 versus 606.1684; the latter values were the average metric values over the other three traditional models, i.e., LR, RF

**Table 3. The ALLMVP framework achieves superior prediction accuracy versus baseline models.**

| Algorithms | Acceptance Ratio | Deviation Ratio | RMSE |
|---|---|---|---|
| LR | 17.14% (6/35) | 0.2559 | 682.1846 |
| A-LR | 40% (14/35) | 0.4389 | 1941.0186 |
| RF | 17.14% (6/35) | 0.1471 | 507.9899 |
| A-RF | 54.29% (19/35) | 0.1976 | 630.5056 |
| LSTM | 11.43% (4/35) | 0.2172 | 628.3306 |
| A-LSTM | 68.57% (24/35) | 0.0732 | 278.7494 |
| LLMVP | 28.57% (10/35) | 0.1325 | 480.1619 |
| ALLMVP | 71.43% (25/35) | 0.037 | 158.2886 |

and LSTM). This evidenced the value of employing LLM in vaccine prediction tasks. Moreover, we also observed that the adaptive value correction mechanism could substantially improve the model accuracy. Specifically, having the adaptive value correction mechanism would increase the acceptance ratios of LR (from 17.14% to 40%), RF (from 17.14% to 54.29%), LSTM (from 11.43% to 68.57%) and LLMVP (from 28.57% to 71.43%) dramatically. It also significantly lowered the deviation ratios and RMSEs of LSTM and LLMVP. This clearly proved the effectiveness of the adaptive value correction mechanism in terms of calibrating model outputs and meeting desired thresholds. Particularly, of all the eight models in comparison, ALLMVP yielded the most accurate predictions, as it had the highest acceptance ratio (71.43%), the lowest deviation ratio (0.037) as well as the lowest RMSE (158.29) among the models. This suggests that the combination of large language model and the adaptive value correction mechanism can mostly enhance the models' predictive capabilities, especially for complex tasks like vaccine demand forecasting.

We further examined the models' predictions for the cumulative five-year demand of each vaccine (S2 Table). The ALLMVP model consistently demonstrated exceptional accuracy across most vaccines. For vaccine A1, the ALLMVP model predicted a cumulative demand of 88,504 doses over the five-year period, closely matching the actual total of 87,150 doses. This resulted in a predicted-to-actual ratio of 1.0155, indicating a minimal deviation of only 1.55%. Similarly, for vaccine A3, the model predicted 92,834 doses against an actual demand of 90,431 doses, yielding a ratio of 1.0266. These results highlight the model's effectiveness in accurately capturing the demand patterns for these vaccines. In contrast, traditional models without Large Language Model (LLM) correction value enhancements, such as LR and LSTM, exhibited significant overestimations. For example, the LR model predicted 100,180 doses for vaccine A1, overestimating the actual demand by approximately 14.95%. The LSTM model showed a similar trend, overestimating the demand for vaccine A1 by 13.27%. This pattern of overestimation was consistent across several vaccines, likely due to the limitations of traditional models in capturing the nuances of vaccine demand. The ALLMVP model also showed strong performance for vaccines A4, A5, A6, and A8. For vaccine A4, the model predicted 46,006 doses compared to the actual total of 44,623 doses, resulting in a ratio of 1.0310—a deviation of only 3.10%. For vaccine A5, the prediction was 160,639 doses against an actual demand of 154,579 doses (ratio of 1.0392). For vaccine A6, the model predicted 129,975 doses compared to the actual 127,518 doses (ratio of 1.0193), indicating high accuracy. For vaccine A8, the ALLMVP model predicted 74,222 doses, closely aligning with the actual demand of 72,213 doses (ratio of 1.0278). However, the ALLMVP model slightly underestimated the cumulative demand for vaccine A7, predicting 79,782 doses against an actual total of 82,119 doses, resulting in a ratio of 0.9715. Although this underestimation is relatively small at 2.85%, it suggests that certain vaccines may present greater forecasting challenges. Potential reasons for this discrepancy could include higher volatility in demand for vaccine A7, irregular vaccination schedules, or external factors affecting vaccine uptake that are not fully captured by the model. For other models, the challenges in predicting certain vaccines were more pronounced. The A-LR model, for instance, significantly overestimated the cumulative demand for vaccine A7, predicting 145,734 doses—an overestimation of approximately 77.47%. Such substantial deviations indicate that while the traditional models can benefit

from the adaptive value correction mechanism, they are still restricted by their capabilities of mining complex demand patterns. Additional data sources or model refinement may be warranted to better capture patterns and factors that drive the demand of the vaccine A7. These findings reinforce the advantage of the ALLMVP model over other models in forecasting vaccine demands.

To further understand model performance before and during the COVID-19 pandemic, we examined annual total vaccine demand predictions for the pre-COVID era (2018–2019) and the COVID era (2020–2022) (S3 Table). For the ALLMVP model, the average deviation ratio prior to COVID-19 was 1.02185, and the average deviation ratio during the pandemic was 1.0211. In contrast, the LR model's average deviation ratio rose from 1.13885 in the pre-COVID era to 1.2239 during the pandemic. Likewise, the performances of most other models were also disrupted by COVID, the average deviation ratios of the A-LR model (1.03655 to 1.1824), the RF model (1.10245 to 1.1624), the A-RF model (1.021 to 1.2084), the LSTM model (1.1102 to 1.2301) and the A-LSTM model (1.0132 to 1.0879) were all significantly increased by COVID disruptions. The LLMVP model, however, displayed an opposite trend that its average deviation ratio was decreased during the pandemic, from 1.15255 to 1.0460. These results suggest that the ALLMVP model performed most stably before and during the COVID pandemic and was little affected by the COVID disruptions observed in most other models.

Irregularities were also observed across multiple vaccines and years, highlighting that some models struggled to predict accurately in scenarios that deviated from typical demand patterns (S1 Table). For example, while discussions often center on well-predicted vaccines like A1(Hepatitis B) or A3(Polio), a closer examination of the evaluation metrics reveals substantial forecasting errors for others. Vaccine A4 (Diphtheria, Tetanus, and Pertussis), known for its volatile demand, faced many challenges in prediction: in 2021, the LR model overestimated A4 demand by more than 70% (ratio 1.7114), and the LSTM model had even larger overestimation at nearly 148%. These results showed that traditional machine learning approaches had difficulty capturing the nuanced trends of vaccines with irregular demand patterns. Additionally, the demand of vaccine A7 (Japanese encephalitis) had even higher volatility that posed challenged for predictions. In 2022, the A-LR model predicted a staggering 66,532 doses for A7 against an actual 11,929, yielding a ratio of 5.5773—far beyond any acceptable margin of error. Other models, such as the A-RF and LSTM models, also encountered troubles for predicting A7 demands in various years. For vaccine A8 (Hepatitis A): although the ALLMVP model had maintained relatively tight deviation ratios for A8 in all years, other algorithms, including those equipped with the adaptive value correction mechanism (like A-RF and A-LSTM), had noticeable deviations. This discrepancy points to a complexity in vaccine A8's demand patterns that not all models were equipped to handle. The aforementioned irregularities suggests that while some models—especially those enhanced with the adaptive value correction algorithms—showed significant improvements and good accuracies for certain vaccines, their did not achieve consistent high accuracies across all vaccine types and in all years. It is those vaccines with irregular/volatile demands that separate the models in terms of their predictive powers and help confirm the stability and accuracy attained by the ALLMVP model. The three evaluation metrics all agree that the ALLMVP model is probably the safest choice for demand forecasting application, among all the models in comparison.

## 4. Discussion

This study investigated a range of forecasting models, both traditional and LLM-enhanced, to predict vaccine demand from 2018 to 2022 in Shanghai, China. Our evaluation metrics—acceptance ratio, deviation ratio, and RMSE—indicate that traditional predictive models such as Linear Regression (LR), Random Forest (RF), and LSTM had difficulty capturing complex temporal patterns, achieving baseline acceptance ratios of only 11% to 17%. By contrast, when large language models (LLMs) were integrated, even basic LLM-enhanced versions significantly outperformed their traditional counterparts, demonstrating that LLMs can more effectively extract patterns from intricate vaccine demand data. For instance, the LLM-based model without the adaptive mechanism (LLMVP) already raised the acceptance ratio to approximately 29%, surpassing that of any traditional model. Building on this LLM foundation, the introduction of an adaptive value correction

mechanism notably boosted the accuracy and reliability of every tested model. Variants such as A-LR, A-RF, and A-LSTM showed dramatic improvements, achieving acceptance ratios of 40%, 54%, and 68% respectively. Most impressively, adapting the specialized LLM-based model (LLMVP) into the ALLMVP framework increased the acceptance ratio further to over 71%. These findings underscore not only the inherent advantage of LLM-based methods over traditional approaches but also the substantial gains that can be realized through adaptive post-processing techniques.

The year-by-year predictions and vaccine-type breakdowns offer additional detail on how adaptive value correction refines forecasting performance. For instance, when examining cumulative five-year totals by vaccine (S2 Table), the ALLMVP model maintained close alignment with actual values across multiple vaccine types—A1(Hepatitis B vaccine) and A3(Polio vaccine) predictions, for example, both remained within about 3% of true totals. In contrast, traditional models (LR, RF and LSTM) typically had substantial biases when predicting the demands of these same vaccines, while their adaptive counterparts (A-LR, A-RF and A-LSTM) had considerably lower biases for the same vaccines, suggesting that the adaptive value correction mechanism help reduce prediction error regardless of the vaccine types. Similarly, when evaluating annual predicted demands over the study period (2018–2022; S3 Table), the ALLMVP model consistently produced ratios closer to 1.0 throughout the entire period, while models without adaptive adjustments drifted more noticeably from actual demands. Long-term aggregated analyses further substantiate this trend: over the full five-year horizon, adaptive approaches not only reduced prediction errors but also ensured more robust performance across varying vaccine types and disruptions (such as COVID). The granular, year-specific and model-specific outcomes evidence the effectiveness of the ALLMVP for forecasting vaccine demands under various complex scenarios.

Furthermore, the ALLMVP model provides concrete guidance for stockpile management. The error analysis for individual vaccines is presented in S4 Table. The model demonstrated high accuracy for vaccines with stable demand patterns, such as the Hepatitis B vaccine (mean absolute percentage error, MAPE = 2.416%), with no significant systematic bias. In contrast, a clear systematic under-forecasting trend was observed for the Japanese encephalitis vaccine (mean bias error, MBE = -467 doses; MAPE = 5.6%), indicating a more challenging demand pattern. Based on these findings, public health program managers could maintain lower safety stock levels for the Hepatitis B vaccine. For the Japanese encephalitis vaccine, however, we recommend adding a buffer stock of approximately 5–10% to the predicted demand to mitigate the risk of shortages. Such differentiated inventory strategies are crucial for optimizing the overall efficiency and resilience of the vaccine supply chain.

From a scholarly perspective, our findings advance the field of vaccine demand prediction by demonstrating that large language models, originally designed for general-purpose language understanding, can be effectively adapted for vaccine prediction tasks (and potentially other specialized prediction tasks). Our results also demonstrate that the adaptive value correction algorithms—particularly when used in large language models (LLMs)—can further reduce bias and meet the requirement about margin of error in forecasts. This approach holds significance not only for vaccine prediction but also for a broader class of applications where predictions are required to be within a margin of error. As AI is growing to be the backbone of healthcare decision-making, we believe the integration of LLMs, adaptive value correction algorithms, as well as deep learning architectures will be the foundation of prediction tools in the next generation [35–37]. Future research may leverage such AI infrastructure to refine prediction applications, through hybrid modeling strategies and generalizable adaptive algorithms across different healthcare domains (beyond vaccine).

## 5. Limitation

This study presents several limitations that warrant consideration:

**Geographical Limitation**: The data for this study were exclusively sourced from Xuhui District, Shanghai. This geographic constraint may limit the generalizability of our findings to other regions, both within China and globally. To enhance the applicability of these findings, future studies should aim to include a more diverse participant pool from various geographic locations [38–42].

**Model Generalization**: The ALLMVP model demonstrated strong performance within the confines of our tests; however, its ability to generalize to other data types or conditions remains untested. The model's performance may vary under different epidemiological scenarios or vaccination strategies [43–45]. Additional studies are necessary to evaluate the efficacy of these models under varying conditions. For example, different vaccine types, vaccination policies, and government requirements may affect the model's accuracy and applicability. Furthermore, its generalizability should be tested for tasks beyond vaccine prediction, ensuring that the model can be reliably extended to other contexts.

**Algorithm-Specific Limitations**: Given the wide range of prediction accuracy among different models (from 11.43% for LSTM to 71.43% for ALLMVP), it's essential to delve deeper into why some models performed better than others [46]. The study could benefit from an analysis that explores the features or conditions under which each model performs best or fails to perform. This will help in understanding model behavior in complex data scenarios.

**Statistical Validation** While the comparative analysis of acceptance and deviation ratios provides substantial evidence of performance differences, the incorporation of formal statistical testing (e.g., to demonstrate the significance of differences between LLM-based and traditional models, or between adaptive and non-adaptive counterparts) and parameter estimation techniques (e.g., to compute confidence intervals for accuracy metrics) would further strengthen the validity and generalizability of the conclusions. Implementing such rigorous statistical analyses represents a valuable direction for future work to provide an even more robust evaluation of the model's predictive strength.

**Irregularities**: The current models may not fully capture irregularities in vaccine supply arising from sudden changes in public health policies or large-scale disruptions. For instance, the COVID-19 pandemic introduced abrupt shifts in vaccine allocation, storage requirements, and administration priorities, which could influence vaccination uptake and effectiveness. Future versions of the model should include mechanisms to adapt dynamically to such unpredictable circumstances, ensuring more robust and reliable forecasting under rapidly evolving conditions.

## 6. Conclusion

This study validates the effectiveness of the Adaptive Large Language Model for Vaccine Prediction (ALLMVP) in predicting vaccine demand based on vaccination record data from Xuhui District, Shanghai. Seven other models were analyzed alongside the ALLMVP model, and we found the additional adaptive value correction mechanism significantly reduced prediction errors for all the machine learning models in comparison, especially the large language model for vaccine prediction (LLMVP). Our results suggest that the infrastructure of the ALLMVP model has great potential in enhancing predictive performance, particularly for vaccine prediction, which is critical for optimizing vaccine deployment and minimizing wastage. Continued development and application of this infrastructure is necessary for large-scale implementations. Future research should explore the generalizability of the ALLMVP model to different geographic regions and maybe other prediction tasks that have basic requirements for margin of error.

## Supporting information

**S1 Appendix. Prompt Engineering Framework for Vaccine Demand Prediction.**
(DOCX)

**S2 Appendix. Parameter Specifications for Predictive Modeling Approaches.**
(DOCX)

**S3 Appendix. Methodology of the Multi-step Monthly Rolling-Forecast Evaluation.**
(DOCX)

**S1 Table. Comparative predictive value of LLMVP and ALLMVP models from 2018 to 2022.**
(DOCX)

**S2 Table. Comparison of Model Predicted Quantities for Each Vaccine from 2018 to 2022.**
(DOCX)

**S3 Table. Comparison of Model Predicted Total Vaccine Quantities from 2018 to 2022.**
(DOCX)

**S4 Table. Summary of forecast error distributions by vaccine type (2018–2022).**
(DOCX)

**S1 Fig. The conceptual architecture of A-LR model.**
(DOCX)

**S2 Fig. The conceptual architecture of A-RF model.**
(DOCX)

**S3 Fig. The conceptual architecture of A-LSTM model.**
(DOCX)

## Author contributions

**Data curation:** Mingming Chen.

**Formal analysis:** Qiangsong Wu.

**Methodology:** Mingming Chen, Zilong Wang, Tenglong Li.

**Software:** Qi Qi.

**Visualization:** Mingming Chen.

**Writing – original draft:** Mingming Chen.

**Writing – review & editing:** Yi Hu, Tenglong Li.

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
