## [Decision Letter · Decision Letter 0]

3 Nov 2025

Response to Reviewers
Revised Manuscript with Track Changes
Manuscript
**Journal Requirements:**

1. Please send a completed 'Competing Interests' statement, including any COIs declared by your co-authors. If you have no competing interests to declare, please state "The authors have declared that no competing interests exist".

2. In the online submission form, you indicated that Due to policy reasons, vaccination data cannot be made public on the Internet. If you need it, you can contact the corresponding author to obtain it.

3. Uploaded as supplementary information.

3. Please provide an Author Summary. This should appear in your manuscript between the Abstract (if applicable) and the Introduction, and should be 150–200 words long. The aim should be to make your findings accessible to a wide audience that includes both scientists and non-scientists. Sample summaries can be found on our website under Submission Guidelines:

https://journals.plos.org/digitalhealth/s/submission-guidelines#loc-parts-of-a-submission

4. Please provide separate figure files in .tif or .eps format.

5. We have noticed that you have uploaded Supporting Information files, but you have not included a list of legends. Please add a full list of legends for your Supporting Information files after the references list.

**Additional Editor Comments (if provided):**

1-Please mention in the discussion how: "statistical testing could be used to demonstrate statistical significance in the performance of LLM models against traditional models, as well as adaptive value-correlated models against their non-adaptive counterpart models. Parameter estimation techniques could also be used to better estimate and generalize the predictive strength of the models." This was a suggestion by a reviewer but is NOT required to for publication. Please include it in the discussion on alternate approaches or future research.

2-Please include a more detailed discussion and literature review on the strength of LLM models in Section 1 could be given in order to provide a more developed reasoning behind the stronger performance of the LLM models against the traditional models as validated in Section 3.

3-Define acronyms on first use; standardize vaccine names across text/tables.

4-Improve some figure captions (axes, units) and consider adding prediction intervals or error bars where appropriate

Please consider these reviewer comments and either revise accordingly or offer a rationale for maintaining the current manuscript:

Suggestions for improvement (minor, actionable)

1) A short example (one vaccine, one year) or pseudocode showing: inputs → prompt template → LLM output → threshold check → adaptive correction will greatly aid reader understanding. Please also list key LLM settings (model/version, temperature, max tokens/top-p) so others can reproduce the behavior.

2) If this acceptance ratio formally reflects “forecast within ±X% of realized demand,” please specify X and provide a brief rationale (e.g., local procurement tolerances). A parenthetical like “25/35 years-by-vaccine cells met the ±5% criterion (71.43%)” would make the number self-explanatory.

3) Confirm time-respecting evaluation and absence of leakage: Providing a single sentence confirming that training uses only past data and that evaluation is strictly out-of-sample by year (or a rolling-origin scheme) would address typical concerns for time-series studies. If you chose simple year-wise holdouts because procurement is planned annually, noting that operational rationale would be helpful.

4) Heterogeneity and practical safeguards: Because volatility usually can differ by vaccine, a small table or figure summarizing per-vaccine error distributions (and any systematic over- or under-forecasting) would be useful to program managers deciding where to add buffers.

**Reviewers' Comments:**

**Comments to the Author**

1. Does this manuscript meet PLOS Digital Health’s publication criteria?

Reviewer #1: Yes

Reviewer #2: Yes

2. Has the statistical analysis been performed appropriately and rigorously?

Reviewer #1: Yes

Reviewer #2: Yes

3. Have the authors made all data underlying the findings in their manuscript fully available (please refer to the Data Availability Statement at the start of the manuscript PDF file)?

Reviewer #1: Yes

Reviewer #2: Yes

4. Is the manuscript presented in an intelligible fashion and written in standard English?

Reviewer #1: Yes

Reviewer #2: Yes

Reviewer #1: Using records from 2014–2022 across seven National Immunization Program (NIP) vaccines in Xuhui District (Shanghai), the study reports improved alignment with realized annual demand and an “acceptance ratio” of 71.43% relative to several comparators. The topic reducing wastage and avoiding stockouts via better annual planning is also timely and meaningful for immunization logistics thus this work is promising and practically relevant. The paper is clearly written, the problem is well motivated, and the multi-vaccine, multi-year framing (including post-COVID) is valuable. However, my suggestions below are light-touch and aimed at improving transparency and reproducibility without asking for extensive new experimentation.

Suggestions for improvement (minor, actionable)

1) A short example (one vaccine, one year) or pseudocode showing: inputs → prompt template → LLM output → threshold check → adaptive correction will greatly aid reader understanding. Please also list key LLM settings (model/version, temperature, max tokens/top-p) so others can reproduce the behavior.

2) If this acceptance ratio formally reflects “forecast within ±X% of realized demand,” please specify X and provide a brief rationale (e.g., local procurement tolerances). A parenthetical like “25/35 years-by-vaccine cells met the ±5% criterion (71.43%)” would make the number self-explanatory.

3) Confirm time-respecting evaluation and absence of leakage: Providing a single sentence confirming that training uses only past data and that evaluation is strictly out-of-sample by year (or a rolling-origin scheme) would address typical concerns for time-series studies. If you chose simple year-wise holdouts because procurement is planned annually, noting that operational rationale would be helpful.

4) Heterogeneity and practical safeguards: Because volatility usually can differ by vaccine, a small table or figure summarizing per-vaccine error distributions (and any systematic over- or under-forecasting) would be useful to program managers deciding where to add buffers.

Additional minor edits:

1)Define acronyms on first use; standardize vaccine names across text/tables.

2) Improve some figure captions (axes, units) and consider adding prediction intervals or error bars where appropriate.

To conclude, this is a significant, practice-oriented contribution. The method is simple, the motivation is strong, and the results are encouraging. With the clarifications above, mainly presentation and reproducibility notes, I believe the paper will be ready for publication.

Reviewer #2: This paper presents an interesting and useful application for AI predictive models in regards to vaccine demand prediction, particularly given the importance of effective resource allocation for public health systems from the onset of the COVID-19 Pandemic. The use of multiple models tested on multiple vaccine types across several years, along with multiple evaluation metrics, indicates robust experimentation and methodological detail.

However, there remain areas for improvement in regards to the comparative analysis of the models's predictive performance in Section 3. While a sufficiently strong amount of data and evidence in regards to the acceptance and deviation ratios of various models is referenced in the evaluation and comparison of the tested models, the analysis currently seems relatively subjective. Statistical testing could be used to demonstrate statistical significance in the performance of LLM models against traditional models, as well as adaptive value-correlated models against their non-adaptive counterpart models. Parameter estimation techniques could also be used to better estimate and generalize the predictive strength of the models.

Furthermore, a more detailed discussion and literature review on the strength of LLM models in Section 1 could be given in order to provide a more developed reasoning behind the stronger performance of the LLM models against the traditional models as validated in Section 3.

**Do you want your identity to be public for this peer review?** For information about this choice, including consent withdrawal, please see our Privacy Policy

Reviewer #1: **Yes:** Precious Esong Sone

Reviewer #2: **Yes:** Shome Chakraborty

**Figure resubmission:**

**Reproducibility:**To enhance the reproducibility of your results, we recommend that authors of applicable studies deposit laboratory protocols in protocols.io, where a protocol can be assigned its own identifier (DOI) such that it can be cited independently in the future. Additionally, PLOS ONE offers an option to publish peer-reviewed clinical study protocols. Read more information on sharing protocols at https://plos.org/protocols?utm_medium=editorial-email&utm_source=authorletters&utm_campaign=protocols

---

## [Decision Letter · Decision Letter 1]

12 Feb 2026

The adaptive large language models for vaccine prediction: A novel approach to vaccine demand prediction with engineered deviation prompts

PDIG-D-25-00475R1

Dear Dr. Li,

We are pleased to inform you that your manuscript 'The adaptive large language models for vaccine prediction: A novel approach to vaccine demand prediction with engineered deviation prompts' has been provisionally accepted for publication in PLOS Digital Health.

Best regards,

Amara Tariq

Section Editor

PLOS Digital Health

**Additional Editor Comments (if provided):**

**Reviewer Comments (if any, and for reference):**

Reviewer's Responses to Questions

**Comments to the Author**

Reviewer #1: All comments have been addressed

Reviewer #2: All comments have been addressed

publication criteria?

Reviewer #1: Yes

Reviewer #2: Yes

3. Has the statistical analysis been performed appropriately and rigorously?

Reviewer #1: Yes

Reviewer #2: Yes

4. Have the authors made all data underlying the findings in their manuscript fully available (please refer to the Data Availability Statement at the start of the manuscript PDF file)?

Reviewer #1: Yes

Reviewer #2: Yes

5. Is the manuscript presented in an intelligible fashion and written in standard English?

Reviewer #1: Yes

Reviewer #2: Yes

Reviewer #1: Thank you for the constructive revision, you have addressed the key points from the first round. The paper now defines the Acceptance Ratio with a clear ±5% tolerance and procurement rationale, consolidates metrics (Acceptance Ratio, Deviation Ratio, RMSE) in a way that makes results immediately interpretable, and clarifies the time-respecting evaluation with an explicit leakage-avoidance statement and a brief forward/rolling description. The description of the “engineered deviation prompts” and the adaptive correction is much clearer with the example/pseudocode and model settings, and the narrative on per-vaccine patterns helps operational readers think about buffers. At this stage my remaining notes are purely presentational and optional (e.g., a compact formula for the Deviation Ratio in Methods, a one-line indication of how often the adaptive correction triggers, and a quick spell-check pass in the appendix). These are editorial in nature and not prerequisites. Overall, the manuscript is clear, reproducible enough to adapt, and useful to public-health planning. I am comfortable recommending acceptance.

Reviewer #2: This paper addresses all comments and concerns regarding its initial submission.

Areas for further statistical analysis and inference is addressed in the "Limitations and Future Work" section.

The literature review is more developed by including a more detailed contextualized discussion regarding the superior predictive performance of LLM models.

The paper provides readers with greater transparency in the study's processes and results. Pseudocode for inputs to outputs methodological pipeline is provided. The time-series nature for the decisions of the predictive models is better explained and visualized. The acceptance ratio threshold is now clearly explained with proper rationale. Captions for charts are revised to be more clearer and descriptive. Error distributions are provided by vaccine for readers to better analyze.

The paper is also amended with greater clarity for readers. Acronyms and abbreviations are clearly defined for keywords. Vaccine names are standardized.

The paper is substantially strengthened to address potential issues for readers from the initial submission. It is now suitable for publication.

**Do you want your identity to be public for this peer review?** For information about this choice, including consent withdrawal, please see our Privacy Policy

Reviewer #1: **Yes:** Precious Esong Sone

Reviewer #2: **Yes:** Shome Chakraborty
